# New Insights of Potassium Sources Impacts as Foliar Application on ‘Canino’ Apricot Fruit Yield, Fruit Anatomy, Quality and Storability

**DOI:** 10.3390/plants10061163

**Published:** 2021-06-08

**Authors:** Sameh K. Okba, Yasser Mazrou, Hayam M. Elmenofy, Ahmed Ezzat, Abdel-Moety Salama

**Affiliations:** 1Deciduous Fruit Department, Horticulture Research Institute, Agricultural Research Center, Giza 12619, Egypt; bahshort@gmail.com; 2Community College, King Khalid University, Abha 62217, Saudi Arabia; ymazrou@kku.edu.sa; 3Department of Agriculture Economic, Faculty of Agriculture, Tanta University, Tanta 31527, Egypt; 4Fruit Handling Department, Horticulture Research Institute, Agricultural Research Center, Giza 12619, Egypt; dr.hmoustafa2015@gmail.com; 5Department of Horticulture, Faculty of Agriculture, Kafrelshaikh University, Kafr El-Shaikh 33516, Egypt; ahmed.kassem@agr.kfs.edu.eg; 6Physiology and Breeding of Horticultural Crops Lab (PBHCL), Faculty of Agriculture, Kafrelsheikh University, Kafr El-Sheikh 33516, Egypt

**Keywords:** K-sources, fruit anatomy, apricot, yield, quality, storability

## Abstract

This is the first report to study the impacts of potassium sources on apricot fruit yield, quality and storability as a preharvest foliar application. Five sources of potassium (K-humate, K-sulphate, K-nitrate, K-silicate and K-citrate), plus water as a control treatment, were applied individually at 0.2% three times on ‘Canino’ apricot over the 2019 and 2020 seasons. The results showed that all potassium salts, applied foliarly, have potential to improve yield, fruit color, and some fruit physical attributes, such as: weight, size and firmness, as well as a reduced lipid peroxidation, accompanied by a low fruit malondialdehyde content reflected in a high tolerance during storage. The K-nitrate treatment was more effective in the improvement of fruit yield, preharvest quality parameters and keeping fruit postharvest quality characteristics from sharp decline during cold storage. Concerning fruit anatomy, K-nitrate and K-citrate showed thicker cuticle and epidermal parenchyma cell diameters, while the K-silicate induced the highest cell wall thickness. K-nitrate was the most economical, and could be recommended for apricot growers in the Nubaria region of Egypt.

## 1. Introduction

The apricot (*Prunus armeniaca* L.) is one of the most important deciduous trees species belonging family Rosaceae [1]. Apricot fruit is highly appreciated by consumers around world due to its flavor, aroma and nutritional value as well as its high concentration of bioactive compounds [2,3]. According to FAO (2020), Mediterranean countries produce more than 60% of total world production. Egypt ranked 10th in terms of production, 1st in term of productivity, while it was 11th in terms of cultivated area across the world in 2019 [4]. There are few numbers of apricot cultivars that exist in Egypt within the warm temperate zone, making it essential to take care of the limited varieties that still exist. This is important in terms of solving problems such as, low fruit yield, poor quality and short storage periods. ‘Canino’ apricot cultivar has a comparative advantage in spite of its late appearance in the Egyptian market (a result of its high need chilling requirements, which are approaching 550 hrs [5]). It shows good performance for both fruit set and yield per tree under Egyptian conditions. However, although this cultivar is characterized by higher fruit size [6], its low soluble sugar fruit content, vulnerability to fruit chilling injury and limited shelf life are considered to be the key problems [7].

Potassium (K) is one of the most important macro-elements, especially for fruit trees, playing an essential role for plant growth and development. Many previous studies have investigated the effects of potassium on the yield and quality of fruit crops [8,9,10,11,12,13]. Potassium as foliar application is an effective, fast and high-efficiency mean to supply the tree with its nutritional needs in critical periods away from soil–cation interactions. This helps to overcome the aforementioned problems inherent to the ‘Canino’ apricot variety [14]. Nevertheless, a high rivalry between new vegetative growth and fruits exists as sink organs, which in turn, negatively impact root growth and K uptake especially, since the soil’s potassium is lower than when in its other forms: fixed and exchangeable [15]. Although potassium is not found in the structure of any organic compound [16], it is the only element that is present in an ionic form in sap plant cells [15]. It has important physiological roles represented in photosynthesis through control of opening and closing stomata, osmoregulation, cell turgor pressure maintenance, sugar translocation via the phloem to fruits, cell division, fruit coloration as well as soluble solids, bioactive K and pigment accumulation in fruits. Additionally, it facilitates enzyme activation, particularly those enzymes that lose their activity at low temperatures and abiotic stress [17,18]. Potassium has commonly been used to boost yield and quality in various crops. Increasing shelf life is also correlated with sufficient K nutrition [19]. However, information about the preferred potassium levels for apricot is thus far limited regarding yield and fruit quality. Additionally, the side effects of anion correlated with potassium requires more studies since some salts may have some harmful effects on human health, especially when applied close to harvest [20]. Fortunately, the fruits of fruit trees have the lowest levels of nitrate accumulation compared to vegetable fruits, roots, tubers and leafy crops [20,21]. 

Previous studies have shown various effects of different potassium forms on yield, fruit quality and postharvest fruit attributes, with contradictory findings according to methods and number of application, in addition to crop type and phenological stages [15,22,23,24,25]. Exogenous application of an appropriate K form could allow for a higher yield, better quality and alleviate chilling injury during cold storage of different fruit species [26]. The effect of K sources might be linked to the application time and rate, as well as fruit species [27]. However, no previous studies compared, among K sources, in which stage and which K source could be applied. Based on the above, the aim of this paper is to study the physiological and anatomical effects of preharvest foliar application of different sources of potassium; potassium humate, potassium sulphate, potassium nitrate, potassium silicate and potassium citrate’s effects on fruit yield, fruit anatomy, quality and the storability of ‘Canino’ apricot trees. 

## 2. Results and Discussion

### 2.1. Fruit Yield 

Figure 1 shows the effect of the studied treatments on the yield of ‘Canino’ apricot trees in the 2019 and 2020 seasons. Significant variation was observed among all evaluated treatments. Trees treated with all potassium sources appeared to have the greatest effect on yield in comparison to the control trees. In this respect, trees treated with either potassium nitrate (KNO_3_) or potassium citrate (K_3_C_6_H_5_O_7_) as a foliar application produced the highest yield in comparison to other K-forms that gave higher values than those of the control trees. This trend was true throughout both studied seasons. Increasing yield of K-treated trees was correlated with an increase in fruit weight [28]. Moreover, the positive effect of K- treatments on yield was reported in the findings of [27], where K-supply was found in a KNO_3_ form as a foliar application, producing significant increase in yield versus the control at 16% higher. The positive effect of potassium citrate is either due to the role of K in photosynthesis and osmoregulation, allowing an import of the assimilates from source to fruits, which in turn leads to increased fruit weight, or, due the role of citric acid in respiration pathways and the production of important energy (ATP synthesis) for all vital reactions inside the cell [29]. Our results support in the results of [30], which reported a 12% and 16% increase in the yield of both of apples and pears in response to foliar potassium application, respectively, when compared to non-sprayed trees (control). One study [31] revealed that all used treatments, including foliar application with potassium citrate, were effective in increasing the yield of ‘Canino’ apricot trees when compared with control. Furthermore, [32] indicated that the foliar application of potassium nitrate and potassium sulphate significantly enhanced the yield of ‘Jaffa’ sweet oranges over the control (water spray). Moreover, the fruit yield of plum trees was improved by foliar application of potassium nitrate as compared to the control [33].

### 2.2. Fruit Weight and Size

‘Canino’ apricot fruit weight and size significantly differed among the tested treatments in the both seasons (Table 1). The maximum weight and size were produced from trees treated by potassium nitrate (KNO3), followed by potassium citrate treatment. Moreover, a general trend was observed that control mostly gave lower fruit weight and size than all potassium applications. An increase in apricot fruit weight and size with potassium application might be due to the fact that potassium plays an important role in translocation of photoassimilates to sink organs (fruits), thus resulting in changes to osmosis inside these organs which allow the continuous import of carbohydrates and water to fruits, and thus, leading to higher fruit weight and size. Another probable explanation for these increases in weight and size is higher fruit water content in the K-treatments. According to Figure 2, it was observed that alteration in fruit water content is closely associated with the fruit’s growth development [34]. These results are in line with the findings of [35], which proved that a higher nectarine fruit weight coincided with more K flow to the fruit. In addition, [27] demonstrated that the pear fruit weight in KNO_3_-applied trees was significantly greater than that observed in the control treatment. Moreover, apple and pear fruit weight were greater in K-foliar treatments [30,36], further confirming our findings, which reported that fruit weight of the ‘Red Delicious’ apple was higher in all used treatments, including potassium salts as well as nitrate and sulphate when compared to non-sprayed trees.

### 2.3. Fruit Water and K Content

The results represented in Figure 2, demonstrate the effect of K-forms on fruit water content. In 2019’s season, there were no significant differences among potassium nitrate, potassium citrate, or potassium humate applications which gave higher percentage of water content when compared to other treatments which had a similar effect. In the 2020 season, apricot fruit tissues from trees receiving K as foliar application generally had higher water content than those taken from control trees which recorded minimum value. Fruits from trees treated with potassium nitrate and potassium citrate forms had the highest percentage of water content. The differences among potassium sulphate, potassium humate, potassium silicate forms did not reach significant levels. Similar results obtained by [37] stated that fruit water content was greater in N and K-fertilized olive trees than control trees, although the significant differences among them were equal. This may attribute to the increase in the proportion of fruit pulp.

Regarding to fruit K concentration, as seen in Figure 3, it was observed that the fruits of non-sprayed apricot trees (control) had the least potassium percentage in comparison to those from trees sprayed with all potassium salts in both 2019 and 2020 seasons. Foliar application with potassium in the form of nitrate or citrate recorded the highest content of K in the fruits in the first season, as well as potassium humate treatment in the second one. Previous studies conducted by [36] on apples proved that potassium salts (KNO_3_, K_2_SO_4_ and KCl) as a foliar application improved fruit K uptake compared to the non- treated trees (control). Moreover, [27] confirmed these results, showing a 43% increase in pear fruit’s K content at maturity in response to foliar K-nitrate treatment when compared to the control.

### 2.4. Fruit Color (L & b Values)

Data in Table 2 revealed that L and b values were significantly higher by the application of all K-forms when compared to control treatment at harvest for both seasons. Regarding L and b after storage, all K-salts recoded the highest value for both seasons. However, the variation among the tested treatments was significant only in the second season. Hence, foliar K applications had a positive effect on the b and L values of fruits, which indicated better coloration in comparison to the control. Similar observations have been reported by [38], who indicated that using potassium as foliar application either in the form sulphate or nitrate enhanced the coloration of pear fruits, especially L and b values, when compared to the control.

### 2.5. Fruit Malondialdehyde (MDA) Content

Lipid peroxidation in cell membranes as result of free radicals (ROS) accumulation in response to different environmental stressors, especially at low temperatures during storage, led to a higher MDA concentration in plant tissues. Therefore, fruit MDA content may be used as an index for chilling injury during cold storage.

The effect of the applied treatments on fruit MDA content, measured after harvest and at the end of the storage period for both studied seasons, are shown in Figure 4. In both seasons, there were significant differences among the tested treatments on MDA content of fruits sampled after harvest. In this respect, the control gave the highest concentration of fruit MDA when compared to other treatments which were equal statistically in the first season, while foliar K-nitrate application only recorded the least value in the second season.

Regarding fruit MDA content measured at end of the storage period, K-salts statistically reduced fruit’s MDA concentrations at the end of storage period in both seasons when compared to control fruits which gave the highest value, followed by potassium citrate treatment. Furthermore, trees treated with potassium silicate or potassium nitrate produced the lowest concentrations of MDA in fruit juice when compared to the rest treatments in both the 2019 and 2020 seasons, with the exception of the potassium humate and potassium sulphate treatments, which had a similar effect.

Briefly, K-treatments statistically reduced fruit MDA content after harvest in the both seasons and after the end of storage period. This pattern may be related to fruit carotene concentration having an antioxidant activity against cellular oxidative stress caused by several ecological stresses such as, drought, low temperature, salinity, as well as sunburn [17]. This explanation agrees with our results in Table 5, whereas the carotenoid concentrations were mostly high in the fruit of trees receiving potassium which was foliarly applied, particularly at harvest, when compared to the control fruits in both studied seasons. In this study, a negative correlation between fruit MDA content and its carotene concentration was observed at harvest, in both seasons, at a level of 0.05, with the value of correlation coefficient being (r)–0.549* and–0.580*, respectively. These conclusions are supported by the results of [39], where it was reported that melon fruit’s MDA concentration was significantly decreased either in foliar or soil applications with potassium silicate compared to control (non-treated).

### 2.6. Weight Loss %

The presented results in Table 3 showed that the loss in fruit’s weight differed significantly among the tested treatments and increased gradually over the storage period. In both seasons, the control recorded higher percentages of weight loss at all assessment days than other treatment. K-applied salts reduced the percentages of weight loss, and values tended to be statistically equal in most cases.

The high loss in weight of control fruits may be due to their high content of MDA (Figure 4), which reflected the high cellular damage resulting from lipid peroxidation in cell membranes [40], subsequently resulting in a loss of the fruit’s water content, leading to fruit weight loss [41]. This conclusion is confirmed by [42] who indicated that Anna apple’s fruit weight loss % was higher in the control treatment than that resulted from trees sprayed with potassium silicate at 0.3%. The interaction effect between preharvest K-treatments and storage periods on fruit weight loss % was significant in both seasons.

### 2.7. Fruit Firmness

Fruit firmness was significantly affected by the studied treatments, through the storage period, accompanied by a decrease in the values with the progress of storage time (Figure 5). Foliar K applications mostly increased fruit firmness when compared with that of the control, except for humate and citrate forms in the first season only. Trees supplied with potassium in either silicate or sulphate forms recorded the highest values, followed by nitrate form. A probable explanation for this observation may be due to an over accumulation of osmolytes and augmented fruit K content (Figure 3) which possibly lead to high pressure potential of mesocarp tissue [43]. These observations agree with the findings of [26], who found that firmness of orange was superior with foliar-applied potassium silicate compared to the control treatment, and [44], who revealed that higher fruit firmness of Amal apricot cultivar was obtained by a higher concentration of potassium silicate than the non-treated trees. Furthermore, [42] stated that the highest fruit firmness of Anna apple was obtained by potassium silicate application at 0.2%. However, the interaction effect between preharvest K-salts treatments and storage periods on fruit firmness was significant in both seasons.

### 2.8. Fruit Total Chlorophyll and Carotene Content

The results in Table 4 demonstrate that the total chlorophyll content of ‘Canino’ apricot fruits, at harvest, through the storage periods as well as its values, decreased over the tested storage periods, and was significantly affected by exogenously K-salts application. In this respect, at harvest, the control gave the highest values (8.065 and 8.228 µg/mL) in the 2019 and 2020 seasons, respectively. Conversely, the potassium citrate and potassium humate treatments mostly recorded the least content of total chlorophyll fruits in both seasons.

Through the storage periods, in both seasons, the K-sulphate treatment gave (7.869 and 7.424 µg/mL) the highest value of total chlorophyll content in fruits, followed by the K-silicate and K-nitrate treatments which were statistically equal, while the degradation in total chlorophyll was higher in the non-sprayed trees than the remaining treatments. The reduction in degradation of the fruit chlorophyll content in comparison to the control was observed in all K-salts applications. The treatments K-sulphate, K-silicate and K-nitrate exceeded expectation, which could be explained by the function of K sources. In one study, [45] indicated that sulphur lead to decrease ACO activity, while, the effect of potassium nitrate contributed to impeding the chlorophyll breakdown, as the nitrate acted as a source of nitrogen [46]. The role of K-silicate was reported by [47] to reduce ethylene production and chlorophyll degradation.

Regarding fruit’s carotenoids content, the results in Table 5 show a gradual increase of in ‘Canino’ apricots with the advancement of the storage period, until the 14th day of cold storage, at which time it decreases. Control fruits recorded the lowest significant change of carotenoids content, while the use of K-sulphate as a preharvest treatment recorded the highest significant concentrations of this parameter followed by the K-nitrate and potassium silicate treatments, without significant differences between these two treatments. Similar results were observed by [48,49], who reported that beta-carotene concentrations were higher in the muskmelon fruits of k-treated plants than in control fruit (non-sprayed trees); the forms of used potassium were K-chloride, K-nitrate, K-sulphate and K-metalosate (glycine amino acid-complexed K). The interaction effect between preharvest K-treatments and storage periods on fruit chlorophyll and carotene content was significant in both seasons.

### 2.9. Fruit SSC and Acidity Content

Regarding the specific effect of various potassium salts on fruit’s SSC content, data of both seasons is displayed in Table 6, and indicates that the differences among the evaluated potassium forms were significant, and values of ‘Canino’ apricot fruit soluble solids showed an increasing tendency through the storage time as previously noted by [50].

At harvest, Fruits from trees that received potassium nitrate as a foliar application recorded the highest fruit SSC %, followed by, in descending order, K-sulphate, K-silicate, K-citrate, K-humate and control. Through different storage periods, it was noticed that high levels of SSC were obtained in fruits treated with preharvest treatments of potassium nitrate (14.801 and 14.869%), followed by potassium humate (13.864 and 13.582%), then potassium citrate (13.248 and 13.021%) in 2019 and 2020 seasons, respectively, while the least value of SSC was detected in the control treatment in both seasons. This may be explained by the potassium level inside the fruits which contributes in translocation photo assimilates and increasing the activity of enzymes responsible for starch hydrolysis to soluble sugars as well as the conversion of starch to sugars. This conclusion finds support in the data presented in Figure 3, wherein a significant positive correlation was also noted between fruit’s K % and fruit SSC content at harvest (r was 0.770 *** and 0.809 *** in the 2019 and 2020 seasons, respectively.) The continuous increase in the values of soluble solids during the different studied storage periods may be due to fruit weight loss. Similar results were seen by [33], who noted that in plums, a higher plum fruit TSS content resulted from potassium nitrate’s application than the control. Also, [27] reported that potassium-complex humic acid, potassium nitrate gave higher values of pear fruit SSC than control.

As for the fruit’s acidity content, significant variations were observed among the all preharvest exogenous applications on fruit acidity values, and these values decreased over the storage periods (Table 7). At harvest in the first season, potassium nitrate treatment and control gave the highest level of fruit acidity, while in the second season the highest acidity was found in the control and potassium silicate. Simultaneously, the potassium citrate consecutively gave the least value (1.509 and 1.371%) in both seasons, respectively.

Through the studied storage periods, the K-salts forms delayed the decrease in the fruit acidity levels compared to the control, with the exception of the forms of citrate and humate in both seasons. Potassium nitrate recorded the highest levels of fruit acidity than other treatments across both seasons, with the exception of the potassium silicate and sulphate treatments in the 2020 season. An increased use of organic acids is likely main reason for decreasing fruit acidity with extended storage time due to the high respiration rate of apricot fruits, which are considered to be a climacteric type.

Generally, all forms of potassium applications resulted in a reduced fruit acidity at harvest in comparison to control, except potassium nitrate in the 2019 season and potassium silicate in the 2020 season. This may attributed to the role of potassium in neutralization acids and reducing fruit acidity [51], while the increase in fruit acidity caused by the potassium nitrate’s application being a source of nitrogen which caused an increase in fruit acidity % as a result of increased synthesis of amino acids, proteins, and other metabolites as well as their subsequent translocation to the fruits [52]. These findings are in conformity with those obtained by [53], who revealed that foliar application with potassium nitrate (250 mg/l) increased acidity in pomegranate fruit juice in comparison to control. Furthermore, [31] in ‘Canino’ apricot trees, they reported that all foliar treatments contained potassium citrate and humic acid and reduced the fruit acidity more than the control.

Regarding the effect of foliar spray treatments on SSC/Acid ratio (Table 8), it was significantly affected in both tested seasons, and its values showed the same trend as SSC through all storage periods. At harvest, a high ratio was found in potassium citrate in the first season, as well as nitrate and sulphate salt in the second one. Meanwhile, control gave the lowest value in both seasons. The highest ratio through the storage periods resulted from potassium humate, followed by potassium citrate in the first season, as well as potassium nitrate in the second one. The least values belonged the control only in the 2019 season, as well as potassium sulphate treatment in the 2020 season. The interaction effect between preharvest K-treatments and storage periods on SSC %, acidity and SSC/Acid ratio were significant in both studied seasons.

### 2.10. Fruit Decay %

Figure 6 illustrates the effect of various forms of potassium on the percentage of fruit decay either after storage at 7 or 21 days during cold storage 0 ± 1 °C, and 90 ± 5% RH plus 2 days at ambient temperature in the 2019 and 2020 seasons. Significant variation was observed among the tested treatments. Decay % in the fruits that were taken out after storage for 7 days and left for 2 days at an ambient temperature, was higher in the control fruits and those harvested from trees treated with potassium humate form than all remaining treatments. Moreover, the potassium forms; silicate, nitrate and citrate had similar effects and gave the least percentage of decay.

Regarding to the fruit decay % after storage at 21 days, plus 2 days on ambient temperature, in most cases, all potassium sources recorded lower percentage of decay than the control treatment which registered the highest value, followed by the potassium humate form. The variation among the remaining potassium salts; citrate, silicate, nitrate as well as sulphate did not reach a significant level. The decrease in the percentage of fruit decay in most potassium treatments which showed significant differences with the control may be due to their effective role in limiting lipid peroxidation by reducing fruit MDA content, either after harvest, or at end of the storage period. This is in agreement with our results, shown in Figure 4.

The positive effect of potassium citrate in decreasing fruit decay % may be attributed to the role of citric acid in minimizing the reduction of total phenolics [41,54] in loquat and longkong fruits, which is considered anti-microbial [55], and additionally its role in suppressing the activities of oxidoreductase enzymes (PPO and POD) which cause the fruit’s surface to become more fragile [41] and susceptible to cold injury, while the positive effect of potassium silicate may be due to the role of silicon in alleviating biotic and abiotic stresses by its deposition between cell wall and cell membrane, keeping barrier against solute outflow, given strength, and rigidity for tissues [56]. The actual effect of potassium nitrate on reducing the percentage of decay in ‘Canino’ apricot fruits may be through several factors; potassium accumulation in tissues in combination with increase the nitrogen uptake and induction of defense genes. Increasing potassium in tissues, which has a good role in disease resistance by enhanced silicification of cell walls [57], or due to its direct role in inhibiting the growth of mycelium of the fungus [58]. Our results in the present study, as shown in Figure 3, also indicated that the K-nitrate and K-citrate forms resulted mostly in higher K-tissues content, with the least values of fruit decay at the same time. The results are in line with the findings of [59] on peaches, which documented the role of citric acid in reducing postharvest fruit decay %. Furthermore, [42] reported that potassium silicate reduced decay % of Anna apple fruits.

### 2.11. Effect of Diverse Potassium Sources as Foliar Application on Fruit Anatomy (Cuticle, Cell Wall Thickness and Epidermal Parenchyma Cell Diameter)

The effect of potassium sources on cuticle, cell wall thickness and epidermal parenchyma cell diameter were presented in Table 9, at harvest; the fruits from trees treated with either K-citrate or K-nitrate had thicker cuticle layer than the rest treatments, followed by the K-silicate form. In terms of the epidermal parenchyma cell diameter, potassium citrate outperformed the control, the K-humate, as well as K-sulphate while at the same time, its effect was comparable to the K-nitrate and K-sulphate salts. Potassium silicate was found to have the thickest cell wall, followed by potassium nitrate and citrate. The cell wall of the control, K-humate and K-sulphate fruits had the same thickness.

Concerning the effect of the tested treatments on these mentioned variables after storage, cold storage caused a decrease trend in the thickness when compared to the values at harvest. In this respect, except for potassium silicate and potassium citrate, the fruits of the K-nitrate treatment resulted in a thicker cuticle layer and parenchyma cell than the other treatments. As for cell wall thickness measured after storage, the control and K-humate fruits had the thinnest cell walls.

The results in Table 9 showed that cold storage had a detrimental impact on parenchymal cells of mesocarp, and this damage was more evident in the parenchyma mesocarp tissue of the control and K-humate fruits than the other treatments (Appendix A). Furthermore, the least affected treatments by cold storage were potassium nitrate, silicate, and citrate which could be attributed to the low(est) MDA levels found in fruits (Figure 4). This damage can be explained by oxidoreductase enzymes which are abundant in parenchymal cells [60] and may interact with substrates to cause browning, which is considered the most important symptom of chilling injury. In line with previously published findings on sweet cherry by [61], who observed the structural changes in sweet cherry pericarp fruit after storage, we found in the control fruit the cell wall of parenchyma mesocarp tissue had hazy boundary and irreversible deformation.

### 2.12. Economic Evaluation of the Tested Treatments

The profitability of the potassium salts applied under this study was analyzed in Table 10 and Table 11, which indicated the cost of the evaluated treatments and the Net income per hectare for each treatment. Maximum profit was achieved by the potassium nitrate treatment (14952.2 US$) (Table 11), although this treatment recorded the highest cost in both seasons.

## 3. Materials and Methods

### 3.1. Tree Materials and Evaluated Treatments

The trial was conducted over two successive growing seasons, 2019 and 2020 on 8-year old ‘Canino’ apricot trees grafted on apricot seedlings in sandy soil (its chemical properties are illustrated in Table 12) under a drip irrigation system at a private orchard located at Elisha village, EL-Nubaria city, Behaira governorate, Egypt. Trees were planted 4 × 5 m apart and received the common horticultural practices usually applied in this region. Thirty six of similar size ‘Canino’ apricot trees were selected and distributed in a randomized complete block design [62] and grouped under six treatments included water as control as well as five foliar K-salts; potassium humate (12.5% K2O), potassium sulphate (50% K2O), potassium nitrate(40% K2O), potassium silicate (15%K2O) and potassium citrate(37% K2O) applied at 0.2% (by 8 L/tree) three times at 7 a.m.; one through stage II (1 May) and two times, at 15 day intervals during stage III of fruit growth (15 May, and 1 June, [63] during 2019 and 2020 seasons.) Two trees were selected as a replicate, for a total of three replicates per treatment.

### 3.2. Measurements and Analysis

At maturity stage (harvest time, at the end of the 1st week of June for both studied seasons) determined based on the fruit color (yellowish green) according to [64]. Fruit samples were selected from each replicate. Samples were directly brought to Physiology & Breeding of Horticultural Crops Lab, Faculty of Agriculture, Kafrelsheikh University, Kafr El-Sheikh, Egypt, cleaned and washed by distilled water, left to dry at room temperature. Infected and all fruits with visual defects were excluded. The samples were divided into two groups: the first one included 10 fruits for each replicate/treatment for determining and measuring the initial quality parameters at harvest. The second one was packed in 40 × 25 × 15 cm carton boxes dimensions. Each treatment consisted of 12 boxes with three replicates and each replicate represented with 4 kg of ‘Canino’ apricot fruits. The boxes were stored at 0 ± 1 °C and 90 ± 5% RH. The variables were recorded at 7 day intervals during storage period.

#### 3.2.1. Tree Yield and Some Fruit Physical Attributes

Average yield per each replicate as kg/tree was estimated. Average fruit weight (g) was calculated by dividing the total weight of 10 fruits on their number. Fruit size (cm^3^) was determined as average of 5 fruits by water displacement. Firmness (Newton (N)/cm^2^) was measured by penetrometer pressure tester (Push-full Dynamometer) equipped with probe (8 mm diameter) for 5 fruits at the equatorial area of both sides of each fruit without removing the peel. The firmness was recorded at harvest as well as at all storage periods.

#### 3.2.2. Fruit K and Water Content

Fruit K and water content were determined at harvest; 50 g of fresh fruit per replicate sliced and dried in an oven at 75 °C for 72 h. The ash was formed at 500 °C, for 8 h, 0.2 g sample was digested with 5 mL of sulphuric acid (conc.), filtered through a filter paper, then diluted to a volume of 50 mL with distilled water and finally fruit K content based on dry weight was determined using flame photometer by [65], while, fruit water content was calculated by the difference between a known fresh fruit weight (initial weight) and the weight that taken after drying at 70 °C till constant weight, and expressed as percentage of the initial fresh weight.

#### 3.2.3. Fruit Malondialdehyde (MDA) Concentration

Fruit malondialdehyde (MDA) content was determined after harvest and at the end of storage; MDA determination based on the method of thiobarbituric acid (TBA) reaction with a slight modification [66,67]. Fresh tissue samples (0.4 g) were homogenized with 20 mL 10% trichloroacetic acid (TCA), followed by centrifugation at 10,000 *g* for 10 min. 2 mL of supernatant (2 mL of 10% TCA used as control) was mixed with 2 mL 0.5% 2-thiobarbituric acid (TBA), heated at 95 °C for 15 min, then centrifuged at 1800 *g* for 10 min after cooling in an ice-water bath to room temperature. The absorbance was read at 450, 532 and 600 nm. The absorbance of the clear solution was measured at 532 nm and corrected for non-specific turbidity by subtracting the absorbance at 600 nm [67]. The amount of accumulated MDA was estimated as follows:MDA (μmol g^−1^ FW) = [6.452 (OD532 − OD600) − 0.559 OD450] × 10 mL/FW.(1)
where: FW was a fresh weight of sample fruit (g).

#### 3.2.4. Fruit Color (L & b Values)

Fruit surface color of 4 sides of 5 individual fruits, after harvest and at the end of storage, was represented by L (luminosity), b (blue- yellow), and was quantified by using Minolta colorimeter (Minolta co. Ltd., Osaka, Japan) according to [68].

#### 3.2.5. Fruit Weight Loss and Decay (DI)

Fruit weight loss % was calculated every seven days after harvest till end of the trial from the following formula:weight loss % = (Weight loss value at the sampling date/initial apricot weight) × 100(2)

DI was visually assessed by evaluating the extent of the microbial decay symptoms on each fruit surface and results were determined using the following scale: 0 = no decay, 1 = less than 1/4 decay, 2 = 1/4–1/2 decay, 3 = 1/2–3/4 decay and 4 = more than 3/4 decay [69] and modified by [70], results were expressed as a percentage and calculated using the following equation:DI = 100 × ∑ [(DI scale) × (number of fruit at the DI scale)]/(5 × total number of fruit).(3)

#### 3.2.6. Fruit Chemical Attributes

##### Fruit Chlorophyll and Carotene Concentrations

Fruit chlorophyll and carotene values were recorded initially and were followed up in seven days intervals throughout the storage period. Fruit chlorophyll and carotene were spectrophotometrically estimated as described by [71], 5 g of fruit was extracted in 30 mL of 80% acetone, left at room temperature in a dark bottle for 24 h., until measuring by spectrophotometer (UV/Visible spectrophotometer Libra SS0PC) and the results were expressed as (µg/mL of the extract) by the following equations:Chlorophyll a (µg/mL) = 12.21 E663 − 2.81 E646(4)
Chlorophyll b (µg/mL) = 20.13 E646 − 5.03 E663(5)
Total chlorophyll (µg/mL) = chlorophyll a + chlorophyll b(6)
Total carotenoids (µg/mL) = [(1000 E470)−(3.27 × chlorophyll a + 104 × chlorophyll b)]/198(7)
where E = Optical density at the indicated spectrum length, b = Fruit SSC and Acidity %.

Through the storage period intervals, fruit SSC % was determined by using Bellingham & Stanley digital refractometer Model: RFM 340-T, while the acidity as a percentage of malic acid was determined by an automatic titration (TitroLine, Model TL 5000, SI Analytics), then SSC\Acid ratio was calculated [72].

#### 3.2.7. Anatomical Studies

For anatomical studies, parts of apricot fruit samples were randomly taken from each treatment, from the fruit equatorial part at harvest and at the end of storage to study the effect of tested treatments on cuticle, cell wall thickness and diameter of ten epidermal parenchyma cells (mesocarp) at harvest, and at the end of the storage period. Samples were fixed in FAA solution (formed of 50% ethyl alcohol, glacial acetic acid and 40% formaldehyde, 90:5:5 as a volumetric ratio) overnight at room temperature, then tissues were dehydrated in a graded alcohol series. The excised tissues were embedded using paraffin wax [73], then were cross sectioned, 12 μm, using a rotary microtome and stained with safranin and fast-green according the method of [74]. The cuticle thickness (CUT), the Epidermal parenchyma cell diameter (EPCD), cell wall thickness (CWT) were investigated microscopically and photomicrographed (Leica, Wetzlar, Germany) [75,76]

### 3.3. Statistical Analysis

The pooled data were statistically analyzed using a randomized complete block design [62]. The data were submitted to the analysis of variance using CoStat software program. The Duncan’s multiple range tests at 5% were used to compare the mean values. Variations at *p* < 0.05 were considered as significant.

## 4. Conclusions

In view of limitations related to the fruits of ‘Canino’ apricot cultivar (low soluble solid content, vulnerability to chilling injury and limited shelf life), the results showed that all potassium salts as a foliar application have high potential to improve fruit yield, fruit color and some physical attributes such as weight, size and firmness. Moreover, reduced lipid peroxidation accompanied by low fruit MDA content reflected a high level of tolerance for damage occurring via abiotic stress caused by cold storage. In this regard, K-nitrate treatment was more effective in improvement of yield, preharvest quality parameters and keeping postharvest fruit quality characteristic from sharp decline during cold storage. Additionally, this treatment was the most economic, and could be recommended for apricot growers in the Nubaria region with its application of 0.2%, 3 times, 1 through stage II (1 May) and two times, at 15 day intervals during stage III of fruit growth (15 May and 1 June). Moreover, further studies should be carried out in the future to determine which salt previously mentioned, and in which stage, should be applied to apricot trees as well as for the other fruit trees. Furthermore, the residual effect of the application of some potassium salts on human health especially when applied close to harvest time could be taken into consideration in future research.

## Figures and Tables

**Figure 1 plants-10-01163-f001:**
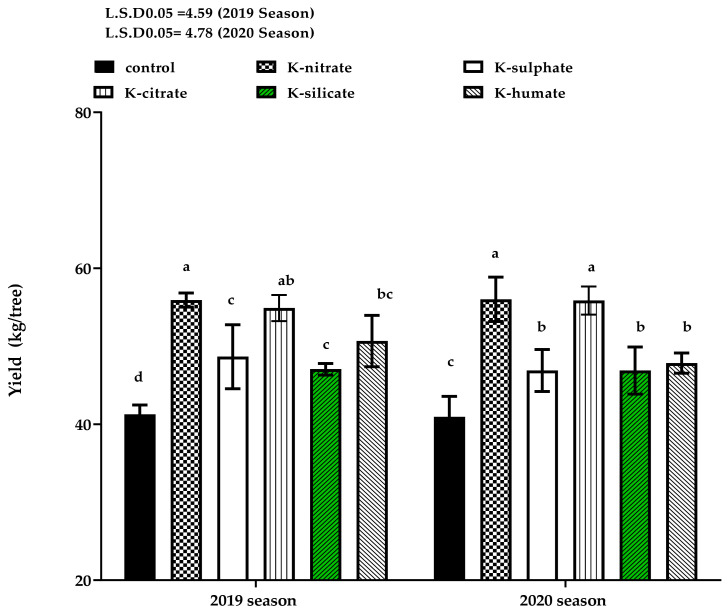
Effect of diverse potassium forms as foliar application on yield of apricot cv. ‘Canino’ in 2019 and 2020 seasons. Data represent the mean with SD. Letters above each bar shown the significant differences among the tested treatment, if letters were same corresponding treatments are statistically equal and vice versa according to Duncan’s multiple range tests at *p* < 0.05.

**Figure 2 plants-10-01163-f002:**
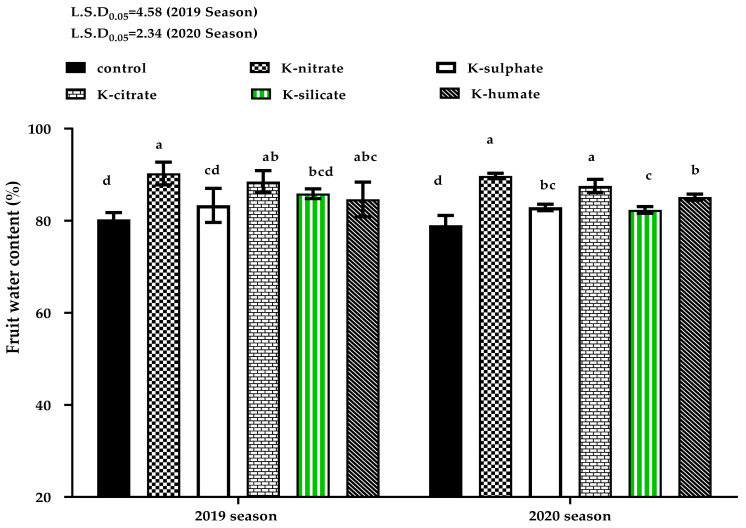
Effect of diverse potassium sources as foliar application on water content of ‘Canino’ apricot fruits in the 2019 and 2020 seasons. Data represent the mean with SD. Letters above each bar shown indicating significant differences among the tested treatment, if letters were same corresponding treatments are statistically equal and vice versa according to Duncan’s multiple range tests at *p* < 0.05.

**Figure 3 plants-10-01163-f003:**
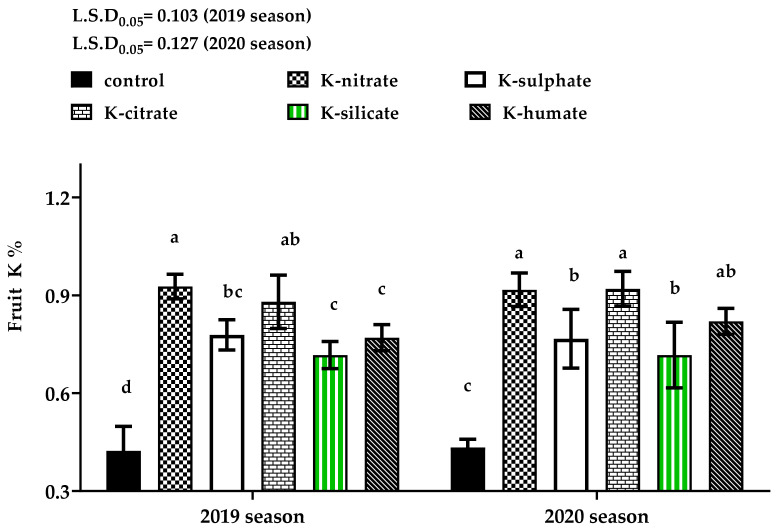
Effect of diverse potassium sources as foliar application on K% of ‘Canino’ apricot fruits in 2019 and 2020 seasons. Data represent the mean with SD. Letters above each bar shown indicating significant differences among the tested treatment, if letters were same corresponding treatments are statistically equal and vice versa according to Duncan’s multiple range tests at *p* < 0.05.

**Figure 4 plants-10-01163-f004:**
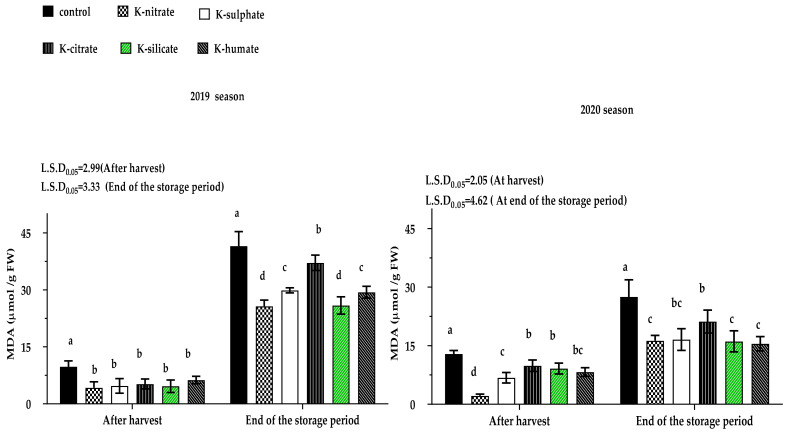
Effect of diverse potassium sources as foliar application on ‘Canino’ apricot fruit MDA content after harvest and at the end of storage period in 2019 and 2020 seasons. Data represent the mean with SD. Letters above each bar shown indicating significant differences among the tested treatment, if letters were same corresponding treatments are statistically equal and vice versa according to Duncan’s multiple range tests at *p* < 0.05.

**Figure 5 plants-10-01163-f005:**
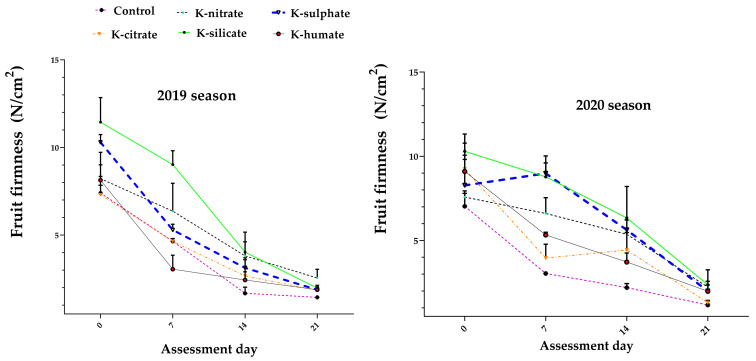
Effect of diverse potassium forms as foliar application on firmness of apricot fruits cv. ‘Canino’ in 2019 and 2020 seasons during cold storage 0 ± 1 °C and 90 ± 5% RH. Values represent the mean with SD.

**Figure 6 plants-10-01163-f006:**
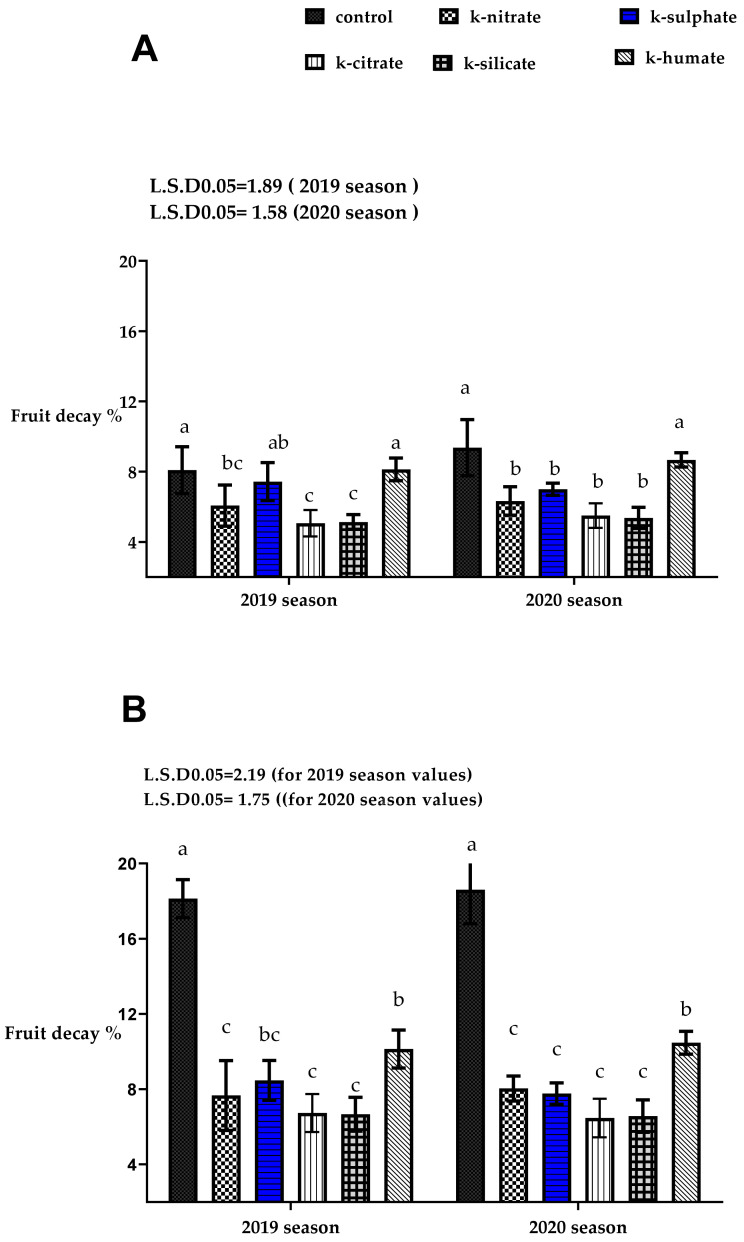
Effect of diverse potassium forms as foliar application on fruit decay % of apricot trees cv. ‘Canino’ (**A**) after storage at 7 plus 2 day on ambient temperature and (**B**) after storage at 21 day plus 2 days at ambient temperature in 2019 and 2020 seasons during cold storage 0 ± 1 °C and 90 ± 5% RH. Values represent the mean with SD. Letters above each bar shown indicating significant differences among the tested treatment, if letters were same corresponding treatments are statistically equal and vice versa according to Duncan’s multiple range tests at *p* < 0.05.

**Table 1 plants-10-01163-t001:** Effect of diverse potassium sources as foliar application on average of fruit weight and size of apricot cv. ‘Canino’ trees in 2019 and 2020 seasons.

Applied Potassium Salts	Fruit Weight (g)		Fruit Size (cm^3^)	
	2019	2020	2019	2020
Control	28.45 c	28.23 d	27.77 e	28.33 d
K-nitrate	43.28 a	42.98 a	42.28 a	42.32 a
K-sulphate	33.56 c	32.33 c	32.70 d	33.33 c
K-citrate	38.62 b	38.94 b	39.23 b	38.51 b
K-silicate	32.44 b	32.34 c	32.34 d	32.67 c
K-humate	35.45 bc	34.67 c	35.67 c	34.92 c

Values within each column followed by different letters are significant at *p* < 0.05 according to the Duncan’s multiple range tests.

**Table 2 plants-10-01163-t002:** Effect of diverse potassium sources as foliar application on color values represented in L and b degree for ‘Canino’ apricot fruits after harvest and at end of the storage in the 2019 and 2020 seasons.

Applied Potassium Salts	2019 Season				2020 Season			
	L *		b *		L *		b *	
	AH ^*^	ES *	AH	ES	AH	ES	AH	ES
Control	55.15 c	49.95 a	33.90 c	31.07 a	58.43 e	45.40 c	34.70 b	26.23 b
K-nitrate	68.10 b	53.50 a	40.90 a	32.47 a	66.23 d	55.40 a	42.40 a	33.53 a
K-sulphate	71.30 b	56.26 a	41.13 a	33.80 a	72.30 bc	47.93 bc	42.75 a	33.20 a
K-citrate	80.57 a	59.10 a	40.33 ab	35.67 a	80.60 a	60.83 a	42.80 a	36.23 a
K-silicate	69.00 b	54.50 a	39.97 ab	33.63 a	67.83 cd	59.00 a	35.93 b	35.53 a
K-humate	65.83 b	52.90 a	37.57 b	31.86 a	77.20 ab	54.30 ab	40.97 a	29.20 b

Values within each column followed by different letters are significant at *p* < 0.05 according to the Duncan’s multiple range tests. L *: (luminosity) & b * (blue–yellow) & AH *: After harvest & ES *: at end of storage.

**Table 3 plants-10-01163-t003:** Effect of diverse potassium sources as foliar application on weight loss % of apricot fruits cv. ‘Canino’ in 2019 and 2020 seasons during cold storage 0 ± 1 °C and 90 ± 5% RH.

Applied Potassium Salts	Fruit Weight Loss %			
	Assessment Day (2019)			
	7	14	21	Mean (A)
Control	6.110	9.366	18.286	8.441 a
K-nitrate	0.663	6.772	9.486	4.231 b
K-sulphate	3.273	4.371	6.028	3.417 bc
K-citrate	2.447	2.901	4.986	2.583 c
K-silicate	1.707	2.808	4.017	2.133 c
K-humate	2.997	6.175	8.582	4.438 b
Mean (B)	2.865 c	5.399 b	8.564 a	
L.S.D at 0.05	Treatments (A) = 1.417 Assessment day (B) = 1.157 Interaction (A × B) = 2.835			
	**Assessment Day (2020)**			
	7	14	21	Mean (A)
Control	5.654	9.767	15.936	7.839 a
K-nitrate	0.906	4.645	6.464	3.003 b
K-sulphate	1.790	4.302	5.694	2.946 b
K-citrate	1.952	3.255	3.989	2.299 b
K-silicate	1.766	2.893	4.453	2.278 b
K-humate	1.804	3.161	6.252	2.804 b
Mean (B)	2.312 c	4.671 b	7.131 a	
L.S.D at 0.05	Treatments (A) = 1.066 Assessment day (B) = 0.871 Interaction (A × B) = 2.132			

The initial weight was 1000 g for each replicate. Values within each column followed by different letters are significant at *p* < 0.05 according to the Duncan’s multiple range tests.

**Table 4 plants-10-01163-t004:** Effect of diverse potassium forms as foliar application on total chlorophyll (µg/mL) of apricot fruits cv. ‘Canino’ in 2019 and 2020 seasons during cold storage 0 ± 1 °C and 90 ± 5% RH.

Applied Potassium Salts	Fruit Total Chlorophyll (µg/mL)				
	Assessment Day (2019)				
	0	7	14	21	Mean (A)
Control	8.065	2.685	1.854	1.839	3.611 e
K-nitrate	6.573	6.482	5.844	6.573	6.368 b
K-sulphate	8.018	7.900	7.708	7.848	7.869 a
K-citrate	5.796	5.781	5.796	3.942	5.329 c
K-silicate	6.852	6.143	5.904	6.727	6.406 b
K-humate	5.440	4.682	5.279	4.044	4.861 d
Mean (B)	6.791 a	5.612 b	5.397 bc	5.162 c	
L.S.D at 0.05	Treatments (A) = 0.306 Assessment day (B) = 0.251 Interaction (A × B) = 0.613				
	**Assessment day (2020)**				
	0	7	14	21	Mean (A)
Control	8.228	2.678	1.740	1.734	3.595 e
K-nitrate	6.573	6.285	5.288	6.414	6.140 b
K-sulphate	7.745	7.410	7.157	7.385	7.424 a
K-citrate	5.551	5.096	5.472	3.106	4.806 c
K-silicate	6.472	6.214	5.898	6.425	6.252 b
K-humate	5.184	4.690	5.126	2.880	4.470 d
Mean (B)	6.625 a	5.396 b	5.113 c	4.657 d	
L.S.D at 0.05	Treatments (A) = 0.275 Assessment day (B) = 0.255 Interaction (A × B) = 0.551				

Values within each column followed by different letters are significant at *p* < 0.05 according to the Duncan’s multiple range tests.

**Table 5 plants-10-01163-t005:** Effect of diverse potassium forms as foliar application on carotene content (µg/mL) of apricot fruits cv. ‘Canino’ in 2019 and 2020 seasons during cold storage 0 ± 1 °C and 90 ± 5% RH.

Applied Potassium Salts	Fruit Carotene Content (µg/mL)				
	Assessment Day (2019)				
	0	7	14	21	Mean (A)
Control	0.844	1.524	1.402	1.183	1.238 c
K-nitrate	1.446	1.606	1.888	1.376	1.579 b
K-sulphate	2.182	2.314	3.219	2.231	2.487 a
K-citrate	1.336	1.300	1.293	1.024	1.238 c
K-silicate	1.358	1.408	2.175	1.359	1.575 b
K-humate	1.391	1.731	1.398	0.440	1.240 c
Mean (B)	1.426 c	1.647 b	1.896 a	1.269 c	
L.S.D at 0.05	Treatments (A) = 0.214 Assessment day (B) = 0.174 Interaction (A × B) = 0.427				
	**Assessment Day (2020)**				
	0	7	14	21	Mean (A)
Control	0.497	1.471	1.419	1.049	1.109 d
K-nitrate	1.356	1.985	2.165	1.440	1.736 b
K-sulphate	2.197	2.434	2.849	2.299	2.445 a
K-citrate	1.344	1.612	1.328	1.009	1.323 c
K-silicate	1.381	1.485	2.008	1.432	1.577 b
K-humate	1.397	1.694	1.374	0.878	1.339 c
Mean (B)	1.362 b	1.780 a	1.857 a	1.351 b	
L.S.D at 0.05	Treatments (A) = 0.203 Assessment day (B) = 0.166 Interaction (A × B) = 0.407				

Values within each column followed by different letters are significant at *p* < 0.05 according to the Duncan’s multiple range tests.

**Table 6 plants-10-01163-t006:** Effect of diverse potassium sources as foliar application on SSC % of apricot fruits cv. ‘Canino’ in 2019 and 2020 seasons during cold storage 0 ± 1 °C and 90 ± 5% RH.

Applied Potassium Salts	Fruit SSC %				
	Assessment Day (2019)				
	0	7	14	21	Mean (A)
Control	8.120	11.170	11.900	14.820	11.503 e
K-nitrate	13.847	14.140	14.657	16.560	14.801 a
K-sulphate	11.693	12.140	14.433	14.550	13.204 c
K-citrate	10.453	13.347	14.280	14.910	13.248 c
K-silicate	11.233	11.900	12.133	13.770	12.259 d
K-humate	9.366	13.760	14.437	17.900	13.864 b
Mean (B)	10.784 d	12.743 c	13.640 b	15.418 a	
L.S.D at 0.05	Treatments (A) = 0.098 Assessment day (B) = 0.079 Interaction (A × B) = 0.195				
	**Assessment day (2020)**				
	0	7	14	21	Mean (A)
Control	7.280	11.190	11.770	14.820	11.265 e
K-nitrate	13.093	13.980	15.183	17.220	14.869 a
K-sulphate	11.600	12.170	13.960	14.167	12.974 c
K-citrate	10.373	13.397	13.823	14.490	13.021 c
K-silicate	11.287	11.980	12.007	13.460	12.184 d
K-humate	9.280	13.570	14.520	16.950	13.582 b
Mean (B)	10.486 d	12.714 c	13.545 b	15.184 a	
L.S.D at 0.05	Treatments (A) = 0.086 Assessment day (B) = 0.070 Interaction (A × B) = 0.172				

Values within each column followed by different letters are significant at P < 0.05 according to the Duncan’s multiple range tests.

**Table 7 plants-10-01163-t007:** Effect of diverse potassium forms as foliar application on acidity % of apricot fruits cv. ‘Canino’ in 2019 and 2020 seasons during cold storage 0 ± 1 °C and 90 ± 5% RH.

Applied Potassium Salts	Fruit Acidity %				
	Assessment Day (2019)				
	0	7	14	21	Mean (A)
Control	2.117	1.306	1.297	1.084	1.451 c
K-nitrate	2.165	1.522	1.503	1.286	1.619 a
K-sulphate	1.905	1.642	1.642	1.587	1.548 b
K-citrate	1.509	1.397	1.189	1.084	1.295 e
K-silicate	1.867	1.555	1.462	1.258	1.536 b
K-humate	1.758	1.696	1.128	0.826	1.352 d
Mean (B)	1.887 a	1.520 b	1.360 c	1.099 d	
L.S.D at 0.05	Treatments (A) = 0.026 Assessment day (B) = 0.021 Interaction (A × B) = 0.052				
	**Assessment day (2020)**				
	0	7	14	21	Mean (A)
Control	1.965	1.286	1.221	0.993	1.366 c
K-nitrate	1.680	1.508	1.414	1.170	1.442 ab
K-sulphate	1.608	1.559	1.465	0.992	1.405 bc
K-citrate	1.371	1.363	1.087	1.083	1.226 d
K-silicate	1.959	1.575	1.359	1.030	1.481 a
K-humate	1.679	1.595	1.079	0.757	1.277 d
Mean (B)	1.711 a	1.481 b	1.271 c	1.004 d	
L.S.D at 0.05	Treatments (A) = 0.065 Assessment day (B) = 0. 054 Interaction (A × B) = 0.132				

Values within each column followed by different letters are significant at *p* < 0.05 according to the Duncan’s multiple range tests.

**Table 8 plants-10-01163-t008:** Effect of diverse potassium forms as foliar application on SSC/Acid ratio of apricot fruits cv. ‘Canino’ in 2019 and 2020 seasons during cold storage 0 ± 1 °C and 90 ± 5% RH.

Applied Potassium Salts	SSC/Acid Ratio				
	Assessment Day (2019)				
	0	7	14	21	Mean (A)
Control	3.836	8.556	9.177	13.677	8.811 e
K-nitrate	6.395	9.293	9.749	12.878	9.579 c
K-sulphate	6.142	7.394	9.098	13.762	9.099 d
K-citrate	6.928	9.553	12.017	13.756	10.563 b
K-silicate	6.018	7.652	8.302	10.945	8.229 f
K-humate	5.326	8.111	12.801	21.674	11.978 a
	5.774 d	8.427 c	10.190 b	14.449 a	
Mean (B)					
L.S.D at 0.05	Treatments (A) = 0.196 Assessment day (B) = 0.160 Interaction (A × B) = 0.392				
	**Assessment Day (2020)**				
	0	7	14	21	Mean (A)
Control	3.705	8.698	9.644	14.930	9.244 cd
K-nitrate	7.792	9.278	10.759	14.754	10.645 b
K-sulphate	7.315	7.857	9.532	14.276	9.745 c
K-citrate	7.565	9.8297	12.723	13.385	10.875 b
K-silicate	5.762	7.729	8.837	13.070	8.849 d
K-humate	5.526	8.511	13.467	22.583	12.522 a
Mean (B)	6.278 d	8.650 c	10.827 b	15.499 a	
L.S.D at 0.05	Treatments (A) = 0.584 Assessment day (B) = 0.485 Interaction (A × B) = 1.169				

Values within each column followed by different letters are significant at *p* < 0.05 according to the Duncan’s multiple range tests.

**Table 9 plants-10-01163-t009:** Effect of diverse potassium sources as foliar application on fruit anatomy (the cuticle thickness (CUT), the Epidermal parenchyma cell diameter (EPCD), cell wall thickness (CWT)) of apricot fruits cv. ‘Canino’ in 2019 and 2020 seasons at harvest and at end of the storage period.

Applied Potassium Salts	μm					
	CUT *		EPCD *		CWT *	
	AH	ES *	AH *	ES *	AH *	ES *
Control	6.86 bc	5.92 b	16.98 c	14.24 c	2.17 c	1.97 c
K-nitrate	11.12 a	9.83 a	24.09 abc	22.80 a	3.09 b	2.58 ab
K-sulphate	5.98 c	5.94 b	16.69 c	15.07 bc	2.75 bc	2.54 ab
K-citrate	11.79 a	8.10 ab	32.01 a	18.71 abc	3.03 b	2.54 ab
K-silicate	8.14 b	8.11 ab	26.29 ab	20.75 ab	3.893 a	2.92 a
K-humate	5.60 c	5.52 b	19.79 c	18.01 abc	2.43 bc	2.30 bc
L.S.D _0.05_	1.24	2.66	7.81	5.38	0.624	0.481

* CUT; The cuticle thickness & EPCD: Epidermal parenchyma cell diameter (mesocarp) & CWT; the cell wall thickness & AH *: After harvest & ES *: at end of storage. Values within each column followed by different letters are significant at *p* < 0.05.

**Table 10 plants-10-01163-t010:** The costs of the treatment of foliar application by potassium salts.

Treatments.	Spraying Rate/ha	Chemical/ha	No. of Applications/ha	Price (US $ /unit)	Total Cost/Chemicals (US $/ha)	Operation Cost (Labors) (US $/ha)	Total Cost (Chemical+ Labors) (US $/ha)
Control							
K-nitrate	1440 L	7.2 kg	3	8.0	58.6	45	103.5
K-sulphate	1440 L	5.76 kg	3	0.5	8.6	45	53.7
K-citrate	1440 L	2.4 L	3	7.08	54	45	99
K-humate	1440 L	2.4 kg	3	3.03	22.6	45	67.5
K-silicate	1440 L	2.4L	3	8.0	58.6	45	103.5

**Table 11 plants-10-01163-t011:** Net income of foliar application of potassium salts.

Treatments	Treatments Cost/ha (US $)	Constant Cost/ha (US $)	Total Cost/ha (US $)	Yield (ton/ha)	Farm Gate Price/ton (US $)	Yield Price (US $/ha)	Net Income/ha (US $)
Control	-	2700	2700	20.790	500	10,395	7695
K-nitrate	103.5	2700	2803.5	28.184	630	17,755.7	14,952.2
K-sulphate	53.7	2700	2753.7	24.525	500	12,262.3	9508.8
K-citrate	99	2700	2799	27.468	500	13,734	10,935
K-silicate	103.5	2700	2803.5	23.708	500	11,854.2	9050.5
K-humate	67.5	2700	2767.5	25.533	500	12,765.9	9998.8

Constant cost includes: 1- Electricity for irrigation = 450 US $/ha 2- Fertilizers (nitrogen + phosphorous + potassium + calcium + trace elements) = 750 US $/ha 3- Pesticides = 750 US $/ha 4-Pruning (summer + winter) = 300,750 US $/ha 5-labors = 450,750 US $/ha 6-Schemes follow the same formatting. US $: US Dollar ha: Hectare.

**Table 12 plants-10-01163-t012:** The chemical properties of the tested soil.

Chemical Properties	EC ds·m^−1^	pH	CaCo_3_ %	Soluble Anions Meq/l				Soluble Cations Meq/l			
				CO_3_^−^	HCO_3_^−^	Cl^−^	So_4_^−−^	K^+^	Mg^++^	Na^+^	Ca^++^
Values	1.45	7.93	8.54	0.00	0.90	0.50	0.26	0.21	0.20	0.45	0.80

## Data Availability

Data is contained within the article and Appendix A.

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
