# Peer review of "New Insights of Potassium Sources Impacts as Foliar Application on ‘Canino’ Apricot Fruit Yield, Fruit Anatomy, Quality and Storability"

_plants, 2021, doi:10.3390/plants10061163_

Round 1
Reviewer 1 Report
The widespread use of foliar sprays for mineral nutrition of fruit trees has so far not been applied with regard to the possible benefits of potassium to apricot quality at harvest and during storage. Herein lies the value of this study. However, the manuscript is too long and should be abbreviated, especially with regard to some of the speculations, which have no data to support them, as to the modes of action of potassium in producing the beneficial effects. For example, the effect of potassium on ethylene biosynthesis and nitric oxide with regard to pigment changes. Another example is the numerous hypotheses as to the effect of potassium on inhibition of decay, where the various salts behave differently. However, the similarity between their differential effects on fruit softening and fruit decay has been overlooked. The deletion of some of the speculations would also enable a shortening of the extremely long list of references, which is not justified.
Specific points:
Tables and Figures
Table numbering: Table 1 appears twice – the first and the last – but there is no Table 2.
It would be easier for the reader if you were to maintain the same order of treatments in all the tables.
Table 4 – the first column is superfluous, as no weight loss can occur on day zero. You might wish to consider replacing it with the initial fruit weight at harvest in order to demonstrate the effect of treatment on fruit weight or that fruit of the same size was used for the test.
Figure 7 – it is difficult to see the differences between the treatments which you describe in the text and as the pertinent data are clearly presented in Table 10, I suggest deleting this figure.
Results and Discussion
Lines 171-183 – As the control does not have an effect, treatments that do not differ from the control also have no effect.
Lines 407-411 – There is no difference in the MDA values between the potassium treatments but there is a difference in the decay values, so this statement is not accurate. The same applies to the potassium content of the treated fruit, which is raised to similar levels.
Materials and Methods:
Line 546 - Was fruit firmness measured on fruit with peel?
Line 574 – Change calorimeter to colorimeter
Line 581 – the DI equation should be division by 4, not by 5. As the data presented in Figure 6 are percentage, do you need to present this equation?
Line 583 – How were the pigments extracted from the fruit?
How many fruit were sampled per replicate for the TSS, acid and pigment analyses?
Author Response
Dear reviewer,
Please see attachment.

Reviewer 2 Report
Dear authors,
changes (lines):
3 ˈCaninoˈ
18-30 Abstract must be rewritten, details of M&M are not part of Abstract
47 to fruit chilling injury
82 ˈCaninoˈ
86 (K3C6H5O7)
87 K-forms
92 k change with K (in all manuscript, 118, 122, 125, 145 etc.)
99 in all manuscript write ˈCaninoˈ (105, 109, 156, 174 etc.)
101 change " Jaffa " in ˈJaffaˈ
110 (Table 1)
145 Figure (3)
165 change l with L
166 Table 2
188 Table 2
230 Table 3
242 Table 3
282 Table 4
307 Table 4
312 Table 5
324 Table 5 and change order of Table 6, 7, 8, 9 and 10
512 Change the order of Discussion and Material & Methods (4. M&M)
527 Soluble anions meq/l – change in Soluble cations meq/l
540 C – change in ⁰C
543 10 fruits per plant is not enough to determine average fruit weight (g), much better method is all fruits from tree and in this case fruits for storage 4 kg (total weight divided with fruit number); exclude this data from results
544-545 for fruit size (cm3), 5 fruits are not enough; it is possible to made big mistakes in measurements
549-554 change k with K
551 missing amount of apricots in grams
574 Minolta calorimeter – change to Minolta colorimeter
577 Weight loss value at the sampling date / initial apricot weight ×100 - change in (Weight loss value at the sampling date / initial apricot weight) ×100
580 Shafiee et al. used different scale
612 with the fruits of ˈCaninoˈ apricot cultivar
613 content
619 post-harvest fruit quality
Please, explain why you used only 10 fruits per plant to determine average fruit weight (g) and only 5 fruits for fruit size (cm3).
It is known that the treatment of fruits with some salts before harvest, especially KNO3, can be a problem for food safety and has potential health risks.
Uddin, R., Thakur, M.U., Uddin, M.Z. et al. Study of nitrate levels in fruits and vegetables to assess the potential health risks in Bangladesh. Sci Rep 11, 4704 (2021). https://doi.org/10.1038/s41598-021-84032-z
Please, explain why you made the treatment with K salts on June 1, when the harvest of the fruit was in the first week of June?
Kind regards
Author Response
Dear reviewer,
Please see attachment.
Best,
Authors

Reviewer 3 Report
Review of the manuscript: New insights of Potassium Sources Impacts as Foliar Application on "Canino" Apricot Fruit Yield, Fruit Anatomy, Quality and Storability
In this study, the impacts of potassium sources on apricot fruit yield, quality and storability as a preharvest foliar application was investigated. The article submitted for review is very important from the point of which treatment is the most economic one and could be recommended for apricot growers at Nubaria region, Egypt and beyond.
Introduction: The introduction is interesting, but in my opinion it does not fully cover the topic. Moreover, out of 27 cited items, one is older than 10 years. The authors refer to some very old literature (item 14). Can the item not be replaced with newer one?
Perhaps the authors will be interested in the article:
Hallmann, E., Rozpara, E., Słowianek, M., & Leszczyńska, J. (2019). The effect of organic and conventional farm management on the allergenic potency and bioactive compounds status of apricots (Prunus armeniaca L.). Food chemistry, 279, 171-178.
This study shows a relationship between the use of organic or conventional practices and the allergenic properties and the bioactive compounds such as carotenoids content of apricots.
Materials and Methods, Fruit chlorophyll and carotene concentrations: It must be pointed that only spectrophotometric method was used to identify the chlorophyll and carotene content of the samples. It is a good way for comparing samples, but not to characterize them. More accurate methods could have been used. It is worth considering the use of not only quantitative but also qualitative analysis. Moreover, the authors refer to very old method (position 35).
It should be noted that the authors described Tree materials and evaluated treatments very thoroughly, as well as the description and graphic representation of the results were presented very well. I believe that the work presented for review is of a high technical level. I am asking for a deeper description, taking into account my suggestions above, with post new items.
Author Response

(The authors gave the same response as above.)

Round 2
Reviewer 2 Report
Dear authors,
you still did not explain why you made the treatment with K salts on June 1, when the harvest of the fruit was in the first week of June. This is so close to pre-harvest treatment with salts and is not in other studies from references.
It is also known that the treatment of fruits with some salts before harvest, especially KNO3, can be a problem for food safety and has potential health risks. Please, note this in your manuscript.
Uddin, R., Thakur, M.U., Uddin, M.Z. et al. Study of nitrate levels in fruits and vegetables to assess the potential health risks in Bangladesh. Sci Rep 11, 4704 (2021). https://doi.org/10.1038/s41598-021-84032-z
You need to improve this manuscript, please check you manuscript for example line 507 Firmness (Lb/inch2 ) must be change in International Metric System (SI) - kg/cm2 or N/cm2.
Regards,
Boris
Author Response
Thank You! Please see attachment.
